# Celiac Disease: Risks of Cross-Contamination and Strategies for Gluten Removal in Food Environments

**DOI:** 10.3390/ijerph21020124

**Published:** 2024-01-24

**Authors:** Fabiana Magnabosco de Vargas, Louise Thomé Cardoso, Amanda Didoné, João P. M. Lima, Janaína Guimarães Venzke, Viviani Ruffo de Oliveira

**Affiliations:** 1Postgraduate Program in Food, Nutrition, and Health (PPGANS), Federal University of Rio Grande do Sul (UFRGS), Porto Alegre 90035-003, RS, Brazil; fabiana.magnabosco@ufrgs.br; 2Postgraduate Program in Agricultural and Environmental Microbiology (PPGMAA), Federal University of Rio Grande do Sul (UFRGS), Porto Alegre 90035-003, RS, Brazil; louise.thome@ufrgs.br; 3Nutrition Deparment, Faculty of Medicine, Federal University of Rio Grande do Sul (UFRGS), Porto Alegre 90035-003, RS, Brazil; amanda.didone@ufrgs.br (A.D.); janaina.venzke@ufrgs.br (J.G.V.); 4Scientific-Pedagogical Unit of Dietetics and Nutrition, Coimbra Health School, Polytechnic University of Coimbra, 3046-854 Coimbra, Portugal; joao.lima@estesc.ipc.pt

**Keywords:** gluten-free diet, gluten removal methods, treatment, gluten removing, gluten residues

## Abstract

Celiac disease (CD) is the chronic immune-mediated enteropathy of the small bowel, manifesting when exposure to gluten occurs in genetically predisposed individuals. Nowadays, the only treatment considered safe for CD is a gluten-free diet (GFD). However, one of the problems faced by celiac patients is the cross-contamination of gluten-free food when preparing meals, in addition to utensils, surfaces and equipment. This study aimed to evaluate cross-contamination in gluten-free products and strategies for removing gluten from cross-contamination in cooking environments. The selection of papers for this integrative review was carried out by searching different databases. Gluten cross-contamination is a global concern for celiac patients in food environments. Although some practices are positive, such as gluten labeling on processed food in several countries, it is crucial to promote good practices in food services around the world. Only a few studies showed effective results in removing gluten from surfaces and utensils; furthermore, sampling was limited, making it difficult to identify appropriate procedures to reduce cross-contamination. The variation in contamination in different kitchen environments also highlighted that celiac patients must continue paying attention to the methods used to prepare gluten-free food. More research is needed, especially into methods of removing gluten from surfaces and utensils, to ensure food safety for celiac patients in many food environments.

## 1. Introduction

Celiac disease (CD) is the chronic immune-mediated enteropathy of the small intestine, which manifests when exposure to dietary gluten occurs in genetically predisposed individuals [1]. The organ most affected by CD is the small intestine, impairing the absorption of nutrients and causing clinical manifestations, which are caused by an inflammatory condition generated by the presence of gluten. Healthy villi are important for the proper digestion and absorption of nutrients [2,3].

Gluten can reach the intestine by some mechanisms: the transcellular route, in which gluten is endocytosed by lysosomes, which degrade it into smaller peptides [4], and the paracellular route, whose tight junctions bind epithelial cells together and promote changes in cell permeability, allowing these gliadin peptides to enter the mucosa. An example is the regulation by zonulin, which is a protein synthesized by the intestine and liver and produced by the epithelial cells of individuals with CD, which causes this change in permeability [2]. Another mechanism is the adaptive immune response, which begins with the exposure of gluten peptides to CD4+ T cells in the intestinal mucosa, leading to the production of proinflammatory cytokines such as interferon-γ (IFN-γ). These cytokines stimulate T helper 1 cells to produce interleukins (IL-15 and IL-21, more specifically), activating CD8+ intraepithelial lymphocytes (IELs) and promoting intestinal damage. In addition, T-helper 2 cells induce the production of antigliadin, anti-transglutaminase and antiendomysial antibodies, characteristic of CD. The increased presence of CD8+ IELs is also an important feature of CD, contributing to inflammation and damage to the intestinal mucosa [5].

After a positive diagnosis, a gluten-free diet is started and an improvement in symptoms is quickly observed. The intestinal mucosa is recovered, in most cases, within two years, and it plays a role in reducing the long-term risk of other complications [6,7]. Gluten and related proteins (e.g., barley and rye) are defined as prolamins, whereas glutelins are seed storage proteins and represent a fraction of glutenin [8]. Gliadins are defined as protein constituents (e.g., wheat flour or gluten proteins) that are insoluble in water or neutral saline solutions but soluble in alcohol. They are rich in proline and glutamine residues that are in polyglutamine sequences. They are classified as monomeric proteins and are either connected to each other through intrachain disulfide bonds (α/β- and γ-gliadins) or not (ω-gliadins) [9,10,11].

So far, the only treatment considered safe for CD is precisely the gluten exclusion diet, and such a diet must be free from gluten cross-contamination [12,13,14], namely, food with up to 20 ppm of gluten is considered safe for people with CD [15]. This silent presence of gluten can cause many problems for people with gluten-related disorders (GRDs) [16].

Cross-contamination can be explained by the occurrence of any food allergen that has not been intentionally added to food as a result of the production, handling, processing, storage, packaging, transportation or preservation of food, or as a result of any environmental contamination [17]. Food cross-contamination is one of the problems faced by patients and can also occur through utensils, surfaces and equipment where gluten has been handled [18], when preparing meals in domestic kitchens or in food services where there has been previous handling of gluten or through food production processes [19]. 

Many celiac patients and other people with GRDs often experience ongoing health challenges caused by a lack of awareness regarding the potential for cross-contamination. This issue is particularly prevalent when shared kitchens are involved, where gluten-containing and gluten-free food items are both processed within the same environment and conventional cleaning practices are employed, as observed by Studerus et al. [12]. Lack of supervision and not applying adequate precautions present a potential risk to the well-being of these individuals.

The viability of gluten removal techniques needs to be further investigated. Since there are few records of techniques already implemented or chemical products used in food service [12,16,20], and given the importance of people who cannot consume gluten to feel safe when eating [19], it is necessary to know the possibilities already evaluated to reduce such contamination in food manipulation.

In this context, the aim of this study was to evaluate cross-contamination in gluten-free products and strategies for removing gluten from the cross-contamination of utensils, surfaces and equipment in cooking environments as preventive measures to minimize or avoid gluten cross-contamination.

## 2. Methods

An integrative review was performed, and papers were selected from 2004 to 2023 by three researchers who followed searching strategies in various databases: Scopus, Science Direct, Web of Science, Springer Link, Gale, Technology Research Database, Cochrane, CAB Direct, PubMed, Lilacs (Latin American and Caribbean Literature in Health Sciences) and Capes Portal. 

The identification and selection of the papers considered the following keywords: “gluten cross-contamination”; “food contamination”; “gluten cross-contact”; “gluten food contamination”; “gluten-free diet”; “proteolytic enzymes”; “gluten residues”; “gluten hydrolysis”; and “equipment contamination”.

The inclusion criteria were (1) compatibility with the main theme; (2) availability for reading; and (3) published articles and gray literature. There were no date limitations, but papers with titles and abstracts outside the topic of interest were excluded. The exclusion criteria were (1) not matching the main theme; (2) duplicates; and (3) studies relating to animals. This established selection criteria found 200 references. Then, 65 were excluded due to the title and/or being duplicates, 49 were excluded after reading the abstracts and more 49 were excluded after reading the full text. Subsequently, the 37 studies (*n* = 34 about cross-contamination in gluten-free products and *n* = 3 about gluten removal methods) were carefully read in their entirety (Figure 1) and included in the study. The data assessed included authors, year of publication, objectives, methods and findings.

## 3. Results and Discussion

### 3.1. Cross-Contamination in Gluten-Free Products

Previous studies were found (*n* = 34) that evaluated cross-contamination in gluten-free products, suggesting that this is indeed a concerning point for people who cannot consume gluten.

Collin et al. [21], in a study conducted in Finland, detected gluten levels of 20–200 ppm in 13 out of 59 gluten-free products and in 11 out of 24 gluten-free products based on wheat starch. Other research has shown that 10–16% of food products apparently commercialized as “gluten-free” in food establishments were contaminated with gluten.

Bustamante et al. [22] in Spain analyzed the evolution between 1998 and 2016 of gluten content in cereal-based gluten-free products using the ELISA technique. The products were split into certified gluten-free products (GF-L) and food claimed to be GF but not certified (GF-NC). Gluten detection has decreased gradually over time, in accordance with the progress of European regulations on food information and gluten content claims. This reduction started earlier for GF-L products than for GF-NC products. Over this time, gluten was detected in 371 samples, with breakfast cereals and cereal bars as the groups most contaminated. Products containing more than 100 ppm of gluten increased in the period of 2013–2016. The data obtained confirm that cereal-based products are improving in safety; however, gluten monitoring must be continued.

To evaluate gluten contamination in Lebanon, 173 samples of gluten-free food were tested over two years. In 6% (*n* = 10) of the samples, the level of gluten was over 20 ppm, and 8 of these contaminated samples were locally produced and were based on wheat starch [23].

In Turkey, a total of 200 samples from eight product categories (snacks, pasta, bread, biscuits, flour and others) made with seven categories of ingredients (cereal mix, buckwheat, corn, rice, carob, potato and others) were analyzed to assess the cross-contamination situation. A high proportion of samples (17.5%) were detected with gluten. Researchers indicated buckwheat as the major cause of this contamination [24].

In India, Raju et al. [25] evaluated the amount of gluten in labels that were naturally gluten-free, such as flour, breakfast products and ready-to-eat food from online grocery stores, supermarkets and local markets, as well as samples of flour obtained straight from the mills, totaling 160 samples. Around 36% of the products were made from gluten-free grains naturally (a mix of gluten-free flour and individual cereal flours, such as rice, oats and millet, as well as legumes, such as lentils, chickpeas and soybeans), and 10% of products labeled as gluten-free (industrialized products based on the flours mentioned above) contained more than 20 ppm of gluten.

A study in Mexico by Calderón de La Barca et al. [26] analyzed products from the northwestern Mexican market labeled as gluten-free and evaluated their gluten content. The study included more than 263 different gluten-free labeled foods, with 55% of them produced in Mexico. Mexican products were mainly flour, sausages, bakery products, dairy products and tortillas, while pasta, snacks and breakfast cereals were mainly imported. Despite 36% of the products being certified, 17.4% of the samples analyzed showed more than 20 ppm of gluten, mostly comprising noncertified products and those made in Mexico. 

In a systematic review carried out in Brazil, Falcomer et al. [27] evaluated studies to estimate cross-contamination with gluten levels above 20 ppm in industrialized food products and observed that 13.2% of such products were above this limit. In studies on nonindustrial food products, an estimate of 41.5% was found.

A study carried out by Siminiuc and Ţurcanu [28] aimed to evaluate the safety of products labeled as gluten-free in the Republic of Moldova for individuals with celiac disease by evaluating their gluten content. The study used the GlutenToxPro gluten detection kit to analyze gluten levels in GF products sold in the state capital’s supermarkets. The findings demonstrated that both products with the Crossed Grain logo and those merely labeled as gluten-free, no matter whether they were imported or produced locally, were safe for those with gluten-related disorders, since their gluten levels were up to 20 ppm. However, locally manufactured, unpackaged GF products available in supermarkets showed a higher risk of contamination.

Mehtab et al. [29] evaluated the presence of gluten in labeled and unlabeled gluten-free food products, as well as imported ones, available in the Indian market. The number of collected products was 794, but 360 were labeled as gluten-free, 80 were imported as gluten-free, and 354 were unlabeled or naturally gluten-free. The amount of gluten was detected using the Ridascreen Gliadin sandwich R5 enzyme-linked immunosorbent assay. In accordance with the standards of the Codex Alimentarius and the Food Safety and Standard Authority of India, gluten-free products should not contain more than 20 ppm of gluten. In general, 10.1% of the GF items tested had gluten content above this level, including 10.8% of the labeled products and 11.8% of the unlabeled/naturally gluten-free products. The imported products were compliant, not containing gluten beyond the recommended limits. The most frequently gluten-contaminated products belonged to categories such as cereals and their derivatives (flour, coarse grains, pasta, snacks), legume flours, seasonings and bakery items. Such results indicate that a high proportion of gluten-free food products available in India exceed the prescribed limits of 20 ppm.

Verma et al. [30] analyzed gluten contamination in gluten-free products available on the Italian market. They collected 200 gluten-free products, including those labeled as natural and certified, from different supermarkets. Using the R5 ELISA Ridascreen^®^ Gliadin sandwich enzyme-linked immunosorbent assay R-7001 (R-Biopharm, Darmstadt, Germany), they determined the gluten levels in each product. The results show that 86.5% of the products contained less than 10 ppm of gluten, 4.5% had between 10 and 20 ppm and 9% had more than 20 ppm. In cases of contamination (gluten > 20 ppm), the amount of gluten was low (between 20 and 100 ppm). The most contaminated foods were oat-based items, buckwheat and lentils. More expensive certified gluten-free products were less likely to contain gluten. The researchers concluded that gluten contamination in gluten-free products marketed in Italy is currently uncommon and generally at low levels. They recommended implementing a systematic sampling program to promptly identify at-risk products.

In the USA, in a pilot study carried out by Thompson, Lee and Grace [31], naturally gluten-free and unlabeled grains, seeds and flour were analyzed. The samples were purchased and submitted for analysis by a specialized gluten deactivation company in 2009. The R5 ELISA Ridascreen^®^ Gliadin sandwich enzyme-linked immunosorbent assay with cocktail extraction was performed in duplicate on all samples. Of the 22 products evaluated, 13 (59%) showed under 5 ppm of gluten, below the quantification level. Nevertheless, nine samples (41%) had gluten levels above the quantification limit, with averages ranging from 8.5 to 2.925.0 ppm. Seven samples (32%) had average gluten levels ≥ 20 ppm, not complying with the criteria of the FDA’s proposed rule for gluten-free labeling. These results highlight the genuine worries about gluten contamination in naturally gluten-free grains, seeds and flour that are commercialized without the “gluten-free” label.

Morais et al. [32] quantified the gluten content in 11 processed foods and verified the accuracy of the labeling about the presence or absence of gluten. This guarantees the availability of quality food for specific population groups, such as individuals with celiac disease, ensuring their right to adequate food. Products were purchased from retail supermarkets in Rio de Janeiro, with selection criteria based on the celiac patients’ diet routines and interests. Approximately 50% of the analyzed food had incorrect information regarding the presence of gluten, which goes against the general rule that food labels must contain clear, accurate and legible information about all components to assist the consumers’ choices according to their needs and dietary restrictions. The researchers emphasize the need to implement corrective actions by health surveillance to ensure that people with celiac disease have access to safe and reliable food, enabling a varied diet suited to their needs.

López et al. [33] investigated possible gluten cross-contamination in Argentina in 37 products containing amaranth, quinoa and/or chia. Using the R5 ELISA Ridascreen^®^ Gliadin sandwich enzyme-linked immunosorbent assay R-7001 (R-Biopharm, Darmstadt, Germany) to detect gliadins (components of gluten), they found that nine samples exceeded the maximum limit of 20 ppm established by the Codex Alimentarius. Surprisingly, some samples were cereal bars labeled “TACC-free” or “gluten-free”, while others were blends of amaranth seeds, cultivars or flours aimed at celiacs. On the other hand, 28 samples presented gluten levels below the established limit, suggesting that food made with amaranth, quinoa and/or chia may be suitable for celiac patients if their production follows good manufacturing practices. However, celiac patients must avoid purchasing such products from retailers, as there is a risk of cross-contamination in these locations.

Guennouni et al. [34] evaluated the gluten content in gluten-free products available in Morocco. A total of 84 food samples were analyzed, including 52 foods labeled as gluten-free and 32 naturally gluten-free foods, distributed across six different categories. Using an enzyme-linked immunosorbent assay (R5 ELISA Ridascreen^®^ Gliadin sandwich—Mendez) with a contamination limit of 20 ppm, the overall contamination rate was 23.8% (21.9% in food labeled gluten-free and 25% in naturally gluten-free food). Of the six categories, three were not contaminated (pasta, cookies and baker’s yeast). The contamination level was 5.3% in dried vegetables, 25% in dried fruit and 42.1% in cereals. All the oat samples were contaminated, and the gluten-free-labeled foods made locally had a higher contamination rate than the imported ones (28.6% vs. 16.7%). These results show a high frequency of gluten contamination in gluten-free products, which affects both labeled and naturally gluten-free food, with oats as the most contaminated one. Consequently, inspections by the competent authorities are needed on a regular basis to avoid the unsafe marketing of these products to celiac patients. Producers should also be strongly encouraged to establish an adequate quality management system.

Gélinas et al. [35] carried out a study in Canada to evaluate gluten contamination in cereal-based food, with or without gluten-free labeling. Of the 148 products tested, around half were labeled as gluten-free. They found, using the R5 enzyme-linked immunosorbent assay (ELI-SA), that 15% of the food had more than 20 ppm. Surprisingly, seven gluten-free foods were contaminated, but they were the least affected. The safest products for celiacs included rice, corn-based food or quinoa. In addition to concerns about misleading labeling, the most critical problems for people with celiac disease (CD) were products made with oats or buckwheat, which were cross-contaminated with wheat and barley gluten. Other problematic products were breakfast cereals, especially those enriched with barley malt ingredients.

Farage et al. [36] analyzed gluten contamination in meals that were naturally gluten-free in food establishments in the Federal District, Brazil. A total of 180 samples of gluten-free meals were selected from 60 places of food service. An immunoenzymatic assay was used to quantify gluten, considering the limit of 20 ppm as acceptable for gluten-free foods. The results indicate that 2.8% of the samples were contaminated by gluten. Among the 60 food services, 6.7% had at least one contaminated food. It is important to emphasize that gluten cross-contamination control is not a consistent practice in Brazil, since their legislation does not demand the prevention of this contamination in food services. In addition, there is no gluten content limit established for foods considered gluten-free in the country.

A study carried out in Italy in a school canteen demonstrated the presence of gluten contamination in supposedly gluten-free meals [37]. Another study carried out in Spain with 50 school canteens found gluten contamination on 83 of the 195 contact surfaces used for gluten-free preparations and, 43 of the 83 were surfaces designated exclusively for the preparation of “gluten-free” meals [38]. The same authors observed that gluten was also the most frequent allergenic residue on the appliances and that a higher number of samples presented high levels of gluten contamination. Gluten, mainly from wheat flour, is easily spread as an aerosol [39,40,41]. Furthermore, such residue is more complex to be eliminated when cleaning because of its low water solubility [20,42].

In a study conducted in a hospital in Croatia [43], researchers developed, implemented and validated a protocol for Hazard Analysis and Critical Control Points (HACCP) to make gluten-free meals in a hospital kitchen, rigorously controlling the amount of gluten. There were no gluten-free food samples from the kitchen, collected up to a year after the application of HACCP, with gluten content above the maximum permitted level of 20 ppm, following European Union standards, demonstrating that the application of the HACCP for the preparation of gluten-free food is useful.

Parsons et al. [44] carried out a study in Utah (United States) in which they tested the cross-contact of gluten-free products after they had been subjected to certain practices likely to cause contamination, such as using a contaminated toaster, the same frying oil used for gluten-containing products and sandwich spreads that have been used before with gluten-containing breads. Each material had 30 measurements taken. Although the authors found gluten even in the gluten-free samples after sharing, such values were below 20 ppm, as established by the Codex Alimentarius, except for “sandwich spreads”. In some samples, the gluten content was above such value, but, on average, they were all below. Researchers understand that there are potential sources of cross-contamination in food products, but it is challenging to identify precisely if this contamination is occurring in the food of people with CD. This investigation has shown that gluten can cross-contaminate many foods inconsistently and is often found in varying amounts.

Similarly, Thompson et al. [45] evaluated gluten-free French fries prepared in fryers shared with wheat products in 10 different restaurants. Of the 20 orders of fries analyzed, 45% had detectable levels of gluten, with 25% of them above the safe limit of 20 ppm. The results indicate that gluten cross-contact can occur in such conditions, and it is recommended that people with celiac disease should avoid foods made in shared fryers.

In Japan, Hashimoto et al. [46] found that washing a bowl previously contaminated with cooked pasta with a sponge and detergent and then using the same sponge to wash a clean bowl, without contact with wheat, transferred wheat allergens with a positive rate of around 80%. They also identified dough traces on the sponge after cleaning and rinsing steps, with a remaining rate of about 20%. A detailed analysis of the residue showed the presence of proteins, especially gluten, which were bonded to the cellular skeleton of the sponge, and among the skeletons, starch granules attached to the proteins. Despite the inclusion of specific sponge washing conditions in the established protocol, achieving complete elimination of wheat allergens proved to be a difficult task. This observation highlights the increased risk of cross-contamination in kitchen establishments devoted to the preparation of allergen-free food when the same sponges are used to clean utensils.

Rostami-Nejad et al. [47] carried out a systematic review and meta-analysis in Iran and found that an intake of 6 ppm/day of gluten was accompanied by a 0.2% chance of CD recurrence, which increased to 1.8% with the consumption of 40 ppm/day of gluten, reaching 50% at the amount of 881 ppm and 100% at the amount of 1500 ppm of gluten per day. The duration of gluten ingestion also showed a positive relationship with CD recurrence, showing that exposure time is as important as quantity. Due to ethical reasons, studies evaluating the response to gluten generally limit the duration of the challenge to short periods, usually three months or less. But, in most cases, the effects of mucosal injury manifest after a longer gluten challenge.

Farage et al. [48] suggested that the ideal would be the consumption of food completely free of gluten, without the possibility of contamination at any stage of production, but the definition of a tolerable value is still under discussion [43], since each patient is unique, and the symptoms of GRD can be very diverse.

Weisbrod et al. [49] carried out five experiments in classrooms to evaluate gluten cross-contamination in activities with modeling dough, bakery dough, *papier-mâché* and dried and fresh pasta. Thirty subjects aged between 2 and 18 were included in the research. After the sessions, gluten levels were tested on slices of GF bread rubbed on the hands and surfaces of the tables used by the participants. The hand-washing method with soap and water was the most effective for removing gluten. *Papier-mâché*, cooked pasta on sensory tables and bakery dough showed high gluten transfer rates above the 20 ppm limit set by the Codex Alimentarius Commission, although modeling dough and dry dough showed low gluten transfer to the GF bread, below this limit.

Oliveira et al. [50] evaluated gluten contamination in beans served in self-service restaurants in Brazil. Around 45% of the restaurants had at least one day of gluten contamination, and 16% of the bean samples were contaminated. This lack of standardization in the preparation of beans represents a huge hazard for people with celiac disease. Public health actions are required to promote safer consumption of gluten-free foods and improve the quality of these individuals’ lives.

Aleksić et al. [51], in a hospital kitchen in the Republic of Serbia, emphasized the importance of applying validated protocols for the cleaning and control of finished products to establish effective control measures for the presence of allergens in food. Such practices are fundamental to ensure good hygiene practices (GHP) and good manufacturing practices (GMP) in hospital facilities. The researchers suggest that, within the scope of food allergen management, it is crucial to verify the effectiveness of cleaning equipment, countertops, employees and the stock of food allergens. Additionally, they recommend that a documented validation method be included in the HACCP plan and periodically reviewed to keep up with dynamic changes in hospital kitchen operations or any changes to business processes, such as the incorporation of new equipment. This is essential to ensure food safety and meet the needs of patients with food allergies.

McIntosh et al. [52] examined CD awareness among food preparation teams in Ireland and assessed their claims of providing gluten-free meals by analyzing a real-time meal sample. Although most attempts to request “gluten-free” meals in restaurants were successful, about 10% of the samples contained gluten (2.7% with levels between 21 and 100 ppm and 7.7% with more than 100 ppm). Surprisingly, two unsatisfactory samples were obtained from self-described “celiac-friendly” restaurants. This indicates that staff confidence and “gluten-free” warnings, signs and menu options were not guarantees of risk-free meals for people with CD. The researchers highlight the need for ongoing training, especially for chefs and restaurant managers, to ensure truly gluten-free and safe meals for individuals with CD. This is essential to guarantee the quality of life and food security of such individuals.

Magalhães et al. [53] evaluated the risks of gluten contamination in a university restaurant that offers meals for celiac patients. It was an observational study carried out in Belo Horizonte, MG, Brazil, between September and November 2014. The researchers used a checklist based on the literature on celiac disease and Resolution #275, of 21 October 2002, to identify critical points of contamination. They found eight critical points with a high risk of contamination, two of which were in the warehouse. Despite the food handlers’ training, the risks of contamination were considered high. Therefore, it is necessary to implement strict control in the production line, separating preparations with and without gluten or using a different physical space to guarantee food safety for celiac patients.

In a study carried out in Italy, in the Piedmont Region, by Bioletti et al. [54], the objective of the investigation was dietary practices in primary and secondary schools under the supervision of SIAN (Food Hygiene and Nutrition Department). The main objective was the analysis of gluten-free meals, encompassing sanitary aspects evaluation and a qualitative assessment of the meals. This assessment was conducted using a previously approved verification protocol. A retrospective analysis of available data was performed to assess the management of gluten-free nutrition in school food services in the Piedmont Region in the year 2010. The study results indicate that 29% of the schools in the sample (a total of 277 institutions) demonstrated compliance with all eight evaluated criteria (including supply, storage, process analysis, equipment verification, packaging and transportation, meal distribution, self-control plan and qualitative evaluation). However, 71% of the assessed schools exhibited inadequacies in at least one of the criteria (with 60% of them not performing the qualitative service evaluation). Additionally, in 18% of the schools, three to seven deficiencies in their dietary practices were identified.

Gluten-free food service products present considerable risks of gluten contamination (Figure 2). Food services should make efforts to minimize the risk of cross-contamination in food, since this would create a more reliable environment for CD patients who need to eat when they are away from home [19]; the procedures to produce gluten-free products should involve good handling and manufacturing practices, effective hygiene methods and training of the work team.

### 3.2. Gluten Removal Methods

Few previous studies (*n* = 3) were found evaluating strategies to minimize gluten removal with different strategies. The proposal of strategies that effectively mitigate gluten and that can be implemented in domestic kitchens, restaurants and industrial kitchens is an aspect that deserves investigation by researchers [20]. To achieve this, it is necessary to know the gluten molecule chemically to be able to remove it from surfaces and utensils so that it is safe to use them for preparations for celiac patients and other GRDs (Table 1). 

Studerus et al. [12] assessed the degree to which gluten cross-contamination occurs through shared kitchen utensils. These researchers evaluated food preparation in a gluten kitchen (kitchen 1) and a gluten-free kitchen (kitchen 2), separated by 4 m. Pasta and bread were prepared in versions with and without gluten. The experiment was carried out as follows: in kitchen 1, pasta with gluten was prepared, which was drained using a stainless-steel pasta colander and a ladle to catch the pasta. Breads were also prepared, and a knife was used to cut them. 

In kitchen 2, gluten-free (GF) pasta and bread were prepared, and the utensils used in kitchen 1 (strainer, knife and ladle) were used in the GF products before being washed (the GF pasta was drained in the contaminated colander and was served with the contaminated ladle, and the GF bread was cut with the contaminated knife). Soon after, four hygiene strategies were carried out to remove gluten, with these utensils being sanitized with cold water, hot water, a clean cloth and towel and with a gluten-contaminated cloth and towel.

The researchers found values below 5 ppm of gliadin (or 10 ppm of gluten) in all analyzed samples in the LFIA tests, except for the ladle, for which a weak signal of gliadin was identified.

Studerus et al. [12] recognize that cross-contamination is a constant concern in the lives of celiac patients, but they suggest that, with appropriate procedures, these individuals can live a more peaceful life, without excessive worry. The authors also recommend paying special attention to toasters, ovens, cutting boards, fryers and utensils that have been shared (for example, with butter, peanut butter or honey, which can be used on gluten-containing bread) because they may present risks of gluten cross-contamination.

Ortiz et al. [38] mentioned in their study that the cleaning process and the effectiveness of detergents still need to be further investigated before being introduced into an allergen cleaning plan. However, it is known that the allergenic potential of wheat gluten can be satisfactorily reduced by enzymatic hydrolysis, since its allergenic epitopes contain between 5 and 20 amino acids [55].

Some enzymes, more specifically proteases, have been used as active elements in detergents to improve their efficiency and effectiveness [56] and even more frequently in the laundry and dishwasher detergent industry [57]. Proteases play a catalytic role in the hydrolysis of peptide bonds and are one of the most important groups of commercial and industrial enzymes [58].

Galan-Malo et al. [16] used detergents with proteases in their study. The samples (stainless steel, Teflon and plastic utensils) were split into two different groups: with or without an additional rinse with detergent-containing proteases. The authors noticed that the use of detergent with proteases significantly reduced the occurrence of allergen residues. In the case of gluten, the decrease was significant and, according to the LFIA findings, the amount of gluten decreased six times.

Fuciños et al. [20] investigated the effectiveness of proteolytic enzymes being added to a standard cleaning product and analyzed their efficiency in removing gluten residues in the food processing industry. The enzymes alcalase, neutrase and flavourzyme (complex of exopeptidases and endo-proteases from *Aspergillus oryzae*) were used. Electrophoretic analyses of the hydrolyzed samples were carried out and confirmed that all enzymes were capable of completely hydrolyzing gluten after 2 h. Alcalase could hydrolyze gluten in just a few minutes, only taking the time to process the sample. This suggests that these enzymes, mainly alcalase and flavourzyme, act very fast, producing small particles that are not detectable by the electrophoresis method. To ensure the effectiveness of gluten removal, it is interesting to confirm it by evaluating gliadin levels.

To find out whether a food or surface is truly gluten-free, immunological methods, such as the Rapid LFIA test [38], and immunoenzymatic ones, such as the ELISA method [59], are used.

## 4. Conclusions

The results show a high level of gluten cross-contamination in gluten-free products.

The findings suggest that cereal-based foods are getting better in terms of safety, although the monitoring of cross-contamination by gluten should be continued.

The few existing studies on this topic do not always show effective results in removing gluten from surfaces and utensils, and sampling was often limited, making it difficult to identify appropriate procedures to reduce cross-contamination. The variation in contamination in different food environments also highlights the need for celiac patients to continue paying attention to the methods used to prepare gluten-free foods.

The cross-contamination of utensils, surfaces and equipment was considered adequately reduced by enzymatic hydrolysis—proteases. Proteases in detergents can also improve their efficiency and effectiveness on kitchen utensils and in the food processing industry.

More studies are needed, especially regarding methods for removing gluten from surfaces and utensils, as the evidence found so far indicates that such elements can be sources of cross-contamination. With more research, it will be possible to develop more precise and effective strategies to ensure food safety for celiac patients in different food environments.

## Figures and Tables

**Figure 1 ijerph-21-00124-f001:**
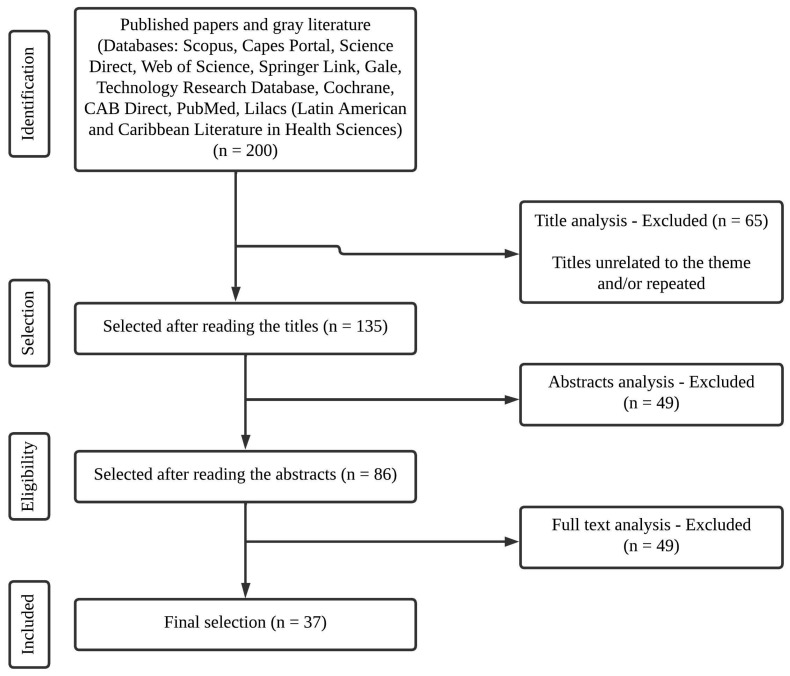
Flowchart for selection, eligibility and inclusion of the articles analyzed in this study.

**Figure 2 ijerph-21-00124-f002:**
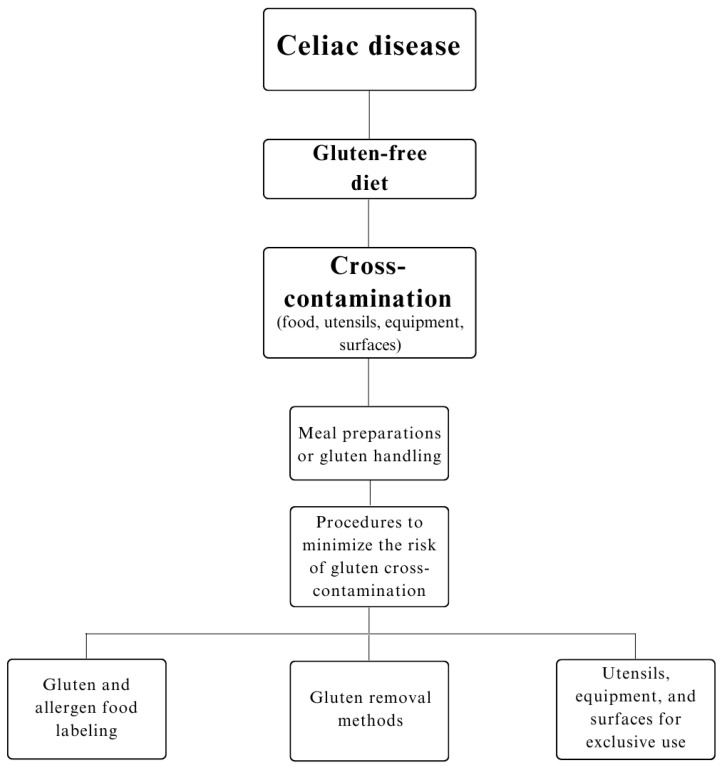
Flow of procedures to minimize the risk of gluten cross-contamination.

**Table 1 ijerph-21-00124-t001:** Methods already evaluated in previous studies to optimize gluten removal.

Author/Year/Country	Utensil/Surface	Gluten Removal Method Used	Analytical Method Used	Effectiveness
Studerus et al. [12] Switzerland	Colander and knife (material: stainless steel)	Washing with cold water, warm water, cleaning with a clean cloth and towel and cleaning with a cloth and towel contaminated with gluten.	ELISA (sandwich) and PCR	<5 ppm gluten
	Ladle	Cleaning with the following: (1) Clean cloth and towel; (2) Cloth and towel contaminated with gluten.		<10 ppm gluten (weak signal)
Galan-Malo et al. [16] Spain	Stainless steel, Teflon and plastic utensils	Cleaning with common detergent	ELISA (sandwich) and LFIA	Small reduction
		Cleaning with protease detergent		Significant reduction
		No extra common washing		Small reduction
		With extra common washing		Significant reduction
		Hand washing		Significant reduction
		Washing in dishwasher		Small reduction
Fuciños et al. [20] Spain	Conveyor belt (plastic)	Enzymatic cleaning for 15 min	ELISA (sandwich and competitive)	<0.125 ppm gluten
	Conveyor belt (stainless steel)	Enzymatic cleaning for 5 min		<0.125 ppm gluten

## Data Availability

Data sharing is not applicable to this article.

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
