# Peer review of "Celiac Disease: Risks of Cross-Contamination and Strategies for Gluten Removal in Food Environments"

_ijerph, 2024, doi:10.3390/ijerph21020124_

Round 1

Reviewer 1 Report

Comments and Suggestions for Authors

This study deals with the contamination of gluten-free foods by residual gluten on surfaces and/or utensils used for their preparation, a topic that is relevant to make every day’s life a lot easier for coeliacs.

However, in spite of the relevance of the topic, a thorough revision of the overall organization of the MS is required in order to transform a disordered report in a focused and organized MS that may be of interest for the reader of this journal.

General remarks:

)     1) The time-coverage of the review is extremely limited (one year!). The time frame must be expanded to include at least the last five years – with improved focus on the title topic – so to provide a more comprehensive and balanced overview of the topic.

2   2) The revised paper should be focused on the title topic, that is, gluten contamination as related to surfaces, technological processes, utensils. Way too many of the papers in the reference list describe the detection of gluten in gluten-free food without paying attention to the nature/origin of contamination. Discussing the many issues involved in procedures for the detection of residual gluten in gluten-free foods is a completely different ballgame, far from the stated aim of this investigation.

Introduction

The introduction in the present version is full of trivial and/or obvious sentences, many of them looking like a cut-and-paste job from other parts of the MS. In this section the Authors should bring forward the practical aspects related to cross contamination from gluten in the environment.

Results

Lines 86-100 are generic sentences and should be moved to the “Introduction”.

The whole sub-section on “comparative legislation” is out of scope and of dubious relevance to the topic. In the present version, the Authors discuss limits for allergens and for gluten in gluten-free foods, making the message quite confusing for the readers. Please note that managing the whole array of food allergens is a completely different story, with little if any relationship to gluten-free foods in terms of both detection of residual amounts of the offending species and of the challenges they represent for food production technologies and processes.

Conclusions

The current version is a collection of trivial statements and should be thoroughly (completely?) rewritten. The Authors should comment on the data appearing under “Results” rather than indulging in what often looks like worthless chit-chat.

Comments on the Quality of English Language

Moderate editing of English language is suggested

Author Response

Dear Editor and Referees,

We would like to thank the referees and the editor for your valuable time during the review process and the insightful comments which helped us to improve the quality of the paper. We have appreciated all the considerations about our paper and we agreed with all of them.

We have added the suggestions and corrections pointed out by the referees. Questions and suggestions are in black and answers are in blue.  If something is not exactly as it was suggested, please let us know and we can write it again.

Referee: 1

Referee(s)' Comments to Author: This study deals with the contamination of gluten-free foods by residual gluten on surfaces and/or utensils used for their preparation, a topic that is relevant to make every day’s life a lot easier for coeliacs.

Response: Thanks for your encouraging statements, we have improved the paper, following our referees’ suggestions. All remarks were accepted and corrected. The authors agree this version is much better than the first one.

However, in spite of the relevance of the topic, a thorough revision of the overall organization of the MS is required in order to transform a disordered report in a focused and organized MS that may be of interest for the reader of this journal.

 Response: We would like to thank our reviewer, who with his/her attentive eye, alerted us that the organization of the paper could be improved.  So, we have modified several parts of the work according to his suggestions. We hope that it is now clearer and easier to read.

General remarks: The time-coverage of the review is extremely limited (one year!). The time frame must be expanded to include at least the last five years – with improved focus on the title topic – so to provide a more comprehensive and balanced overview of the topic.

 Response: We apologize to our reviewer if in the first version the way it was written was misunderstood, we improved this part of the paper as well. In fact, the review covered the last 20 years, as you can confirm the references and table 1, in fact from 2022 to 2023 was the period that the study was carried out. 

 These studies below for example are from previous than 2022 and 2023.

 Studerus et al.  [8] Switzerland, 2018.

 Fuciños et al.  [12] Spain, 2019.

 Galan-Malo et al. [13] Spain, 2019.

   2) The revised paper should be focused on the title topic, that is, gluten contamination as related to surfaces, technological processes, utensils. Way too many of the papers in the reference list describe the detection of gluten in gluten-free food without paying attention to the nature/origin of contamination. Discussing the many issues involved in procedures for the detection of residual gluten in gluten-free foods is a completely different ballgame, far from the stated aim of this investigation.

 Response: Thank you for your suggestion. The part about cross-contamination in gluten-free products is important, in fact it shows the reality that is experienced and neglected, we would like to keep it. But according to your advice we added it in the title and objective. We hope we have improved to make it clearer and to provide a more comprehensive and balanced overview of the subject in this 2nd version. You are completely right, it was weak.

Introduction

The introduction in the present version is full of trivial and/or obvious sentences, many of them looking like a cut-and-paste job from other parts of the MS. In this section the Authors should bring forward the practical aspects related to cross contamination from gluten in the environment.

Response: The introduction has been reformulated, since some topics was requested by the other reviewers as well. We followed your suggestion to bring forward the practical aspects related to cross contamination from gluten in the environment.

 Results

Lines 86-100 are generic sentences and should be moved to the “Introduction”.

Response: Thank you for your advice. We have corrected it in this new version as proposed by our referee. We agree that it is much better.

The whole sub-section on “comparative legislation” is out of scope and of dubious relevance to the topic. In the present version, the Authors discuss limits for allergens and for gluten in gluten-free foods, making the message quite confusing for the readers. Please note that managing the whole array of food allergens is a completely different story, with little if any relationship to gluten-free foods in terms of both detection of residual amounts of the offending species and of the challenges they represent for food production technologies and processes.

 Response: We have removed this section following the recommendations of our reviewer.

Conclusions

The current version is a collection of trivial statements and should be thoroughly (completely?) rewritten. The Authors should comment on the data appearing under “Results” rather than indulging in what often looks like worthless chit-chat.

Response: We have rewritten the conclusion, we agree with you, several important points were not covered in the 1st version. We have removed two paragraphs that might have seemed like "chit-chat" and we have been more assertive. We thank our reviewer for pushing us.

Reviewer 2 Report

Comments and Suggestions for Authors

This review from Magnabosco de Vargas et al is an overview of recent studies addressing the risk of gluten cross-contamination in food environments that represents a serious concern for the community of celiac patients worldwide. The authors accurately describe existing strategies to remove or reduce gluten amounts due to cross-contamination in food management both at familiar, restaurant or industrial levels.

The selection of papers was carried among those published in one year (June 2022 and June 2023) and keywords and inclusion criteria were both well designed. Overall, the authors have examined different studies. The review topic is very timely and of interest, the manuscript is fairly well written and structured.

I have a few comments that need to be addressed by the authors before an overall positive evaluation.

Introduction:

The sentence “People with CD produce antibodies that, when combined with cytokines, can be directly influenced by immune cells, with consequent damage to the intestine and villi flattening” is obscure and should be rephrased. There is no influence of antibodies and cytokines on CD pathogenic T cells.

The definition of gluten given needs a correction. Gluten and related proteins of barley and rye are defined as prolamins, the glutelin are seed storage proteins with high molecular weight, corresponding to a glutenin fraction. The Authors correctly quote

the review from Shewry, P. (Ref 6), but Verdu, E. F., & Schuppan, D. (Ref 7) focus on clinical and pathogenic mechanisms, I suggest to remove it.

Results:

This sentence “Nevertheless, nine samples (41%) had gluten levels above the quantification limit, with averages  ranging from 8.5 to 2,925.0 ppm” is unclear, please clarify what it means the averages ranging from 8.5 to 2,925.0 ppm.

Page 5 line 206, the gluten amount description has to be uniformed, use 20 mg/kg or 20 ppm throughout the text.

Again, the R5 ELISA Ridascreen and R5 ELISA Ridascreen ®gliadin Mendez has to be uniformed throughout the text.

Page 11 line 433. The sentence: Gliadins are defined as protein constituents of wheat flour or gluten proteins that are insoluble in water or neutral saline solutions, but soluble in alcohol. They  are rich in proline and glutamine residues that are in polyglutamine sequences. They are classified as monomeric proteins and are either connected to each other through intra-chain disulfide bonds (α/β- and γ-gliadins) or not (ω-gliadins).” has to be moved in the introduction, more useful as background of this review.

What the authors meant in the following sentence “ The development of protocols for effective gluten elimination is one of the most critical aspects of any allergen management strategy, both in domestic and industrial kitchens” is not clear. The removal of any risk for cross-contamination from utensils and surface is common to all allergens.

Conclusions:

The conclusions are too succinct, it would be worthy to recall the main findings shown in the results and expand the Authors’ side opinions on the topic.

Author Response

Dear Editor and Referees,

We would like to thank the referees and the editor for your valuable time during the review process and the insightful comments which helped us to improve the quality of the paper. We have appreciated all the considerations about our paper and we agreed with all of them.

We have added the suggestions and corrections pointed out by the referees. Questions and suggestions are in black and answers are in blue.  If something is not exactly as it was suggested, please let us know and we can write it again.

 Referee: 2

 Referee(s)' Comments to Author:

This review from Magnabosco de Vargas et al is an overview of recent studies addressing the risk of gluten cross-contamination in food environments that represents a serious concern for the community of celiac patients worldwide. The authors accurately describe existing strategies to remove or reduce gluten amounts due to cross-contamination in food management both at familiar, restaurant or industrial levels.  The selection of papers was carried among those published in one year (June 2022 and June 2023) and keywords and inclusion criteria were both well designed. Overall, the authors have examined different studies. The review topic is very timely and of interest, the manuscript is fairly well written and structured.

 I have a few comments that need to be addressed by the authors before an overall positive evaluation.

Response: Thanks for your kind and encouraging words, we have improved the paper, following our referees’ suggestions. All observations were accepted and corrected, besides all the authors agreed this version is much better than the first one.

Introduction:

The sentence “People with CD produce antibodies that, when combined with cytokines, can be directly influenced by immune cells, with consequent damage to the intestine and villi flattening” is obscure and should be rephrased. There is no influence of antibodies and cytokines on CD pathogenic T cells.

 Response: The introduction has been reformulated, since some topics was requested by the other reviewers as well. We followed your suggestion and corrected this sentence.

The definition of gluten given needs a correction. Gluten and related proteins of barley and rye are defined as prolamins, the glutelin are seed storage proteins with high molecular weight, corresponding to a glutenin fraction. The Authors correctly quote the review from Shewry, P. (Ref 6), but Verdu, E. F., & Schuppan, D. (Ref 7) focus on clinical and pathogenic mechanisms, I suggest to remove it.

 Response: We would like to thank our reviewer, who with his/her attentive eyes, alerted us that the sentence could be improved.  We hope that it is now clearer and easier to read. The new sentence is “Gluten and related proteins (e.g. barley and rye) are defined as prolamins, whereas glutelins are seed storage proteins and represent a fraction of glutenin.”

 Results:

 This sentence “Nevertheless, nine samples (41%) had gluten levels above the quantification limit, with averages ranging from 8.5 to 2,925.0 ppm” is unclear, please clarify what it means the averages ranging from 8.5 to 2,925.0 ppm.

 Response: In the original article (below) you can check that it is exactly like that, we completely agree with you, but unfortunately, the problems with gluten dispersion are huge. We have verified a similar situation in another study of our research group, but it is still in progress and has not been published.

Thompson, T., Lee, A. R., & Grace, T. (2010). Gluten contamination of grains, seeds, and flours in the United States: a pilot study. Journal of the American Dietetic Association, 110(6), 937-940.

Page 5 line 206, the gluten amount description has to be uniformed, use 20 mg/kg or 20 ppm throughout the text.

Response: It was corrected as our referee has suggested.

Again, the R5 ELISA Ridascreen and R5 ELISA Ridascreen ®gliadin Mendez has to be uniformed throughout the text.

Response: Thank you for your suggestion.

We checked all the papers, two kits were used. We have inserted the requested details in the paper.

R5 ELISA Ridascreen® Gliadin sandwich - Mendez

R5 ELISA Ridascreen® Gliadin sandwich, enzyme-linked immunosorbent assay R-7001 (R-Biopharm, Darmstadt, Germany)

Page 11 line 433. The sentence: Gliadins are defined as protein constituents of wheat flour or gluten proteins that are insoluble in water or neutral saline solutions, but soluble in alcohol. They  are rich in proline and glutamine residues that are in polyglutamine sequences. They are classified as monomeric proteins and are either connected to each other through intra-chain disulfide bonds (α/β- and γ-gliadins) or not (ω-gliadins).” has to be moved in the introduction, more useful as background of this review.

Response: The sentence was  moved to the introduction as our referee suggested.

 What the authors meant in the following sentence “The development of protocols for effective gluten elimination is one of the most critical aspects of any allergen management strategy, both in domestic and industrial kitchens” is not clear. The removal of any risk for cross-contamination from utensils and surface is common to all allergens.

Response: The sentence was rephased. We agree, it was quite confused. We hope is clearer now. The proposal of strategies that effectively mitigate gluten is an aspect that deserves investigation by researchers, and that can be implemented in domestic kitchens, restaurants, and industrial kitchens.

Conclusions:

 The conclusions are too succinct, it would be worthy to recall the main findings shown in the results and expand the Authors’ side opinions on the topic.                                  

Response: Thanks for your advice.  We have rewritten the conclusion, we agree with you, several important points were not covered in the 1st version. We have removed two paragraphs included some new assertive points. We thank our reviewer for pushing us.

Reviewer 3 Report

Comments and Suggestions for Authors

Celiac Disease (CD) is a chronic immune-mediated enteropathy of the small bowel, manifested when exposure to dietary gluten occurs in genetically predisposed individuals. Now, the only treatment considered safe for CD is the gluten-free diet (GFD). However, one of the problems faced by celiac patients is cross-contamination of food by utensils, surfaces, and equipment when preparing meals. To minimize this risk, effective control measures can be adopted, such as good handling and manufacturing practices, following legislation, and training the work team. This paper evaluates existing strategies as preventive measures in removing or reducing gluten amounts due to cross-contamination through an integrative review. The selection of papers was carried out by searching databases between June 2022 and June 2023. Gluten cross-contamination is a global concern for celiac patients in food environments. Although some practices are positive, such as gluten labeling on processed foods in several countries, it is crucial to promote good practices in food services around the world. There are only few studies which did not show effective results in removing gluten from surfaces and utensils, besides sampling was limited, making it difficult to identify appropriate procedures to reduce cross-contamination. The variation in contamination in different kitchen environments also highlighted that celiac patients must continue paying attention to the methods used to prepare gluten-free foods. More research is needed, especially into methods of removing gluten from surfaces and utensils, to ensure food safety for celiac patients in many food environments.

The quality of this manuscript is good. It is well design and the result is interesting. However, there still have some issues need to improve.

1.      The Celiac Disease with inflammation should be analyzed.

2.      The gluten-free diet and Celiac Disease should be analyzed. Please refer this reference (Journal of Food Processing and Preservation,2021, 45(9),e15684.).

3.      The cytokines or inflammation status should be evaluated.

4.      The reason for combination and dose basis.

5.      The expression should be double checked.

6.      The reference should be updated in recent years.

Comments on the Quality of English Language

Celiac Disease (CD) is a chronic immune-mediated enteropathy of the small bowel, manifested when exposure to dietary gluten occurs in genetically predisposed individuals. Now, the only treatment considered safe for CD is the gluten-free diet (GFD). However, one of the problems faced by celiac patients is cross-contamination of food by utensils, surfaces, and equipment when preparing meals. To minimize this risk, effective control measures can be adopted, such as good handling and manufacturing practices, following legislation, and training the work team. This paper evaluates existing strategies as preventive measures in removing or reducing gluten amounts due to cross-contamination through an integrative review. The selection of papers was carried out by searching databases between June 2022 and June 2023. Gluten cross-contamination is a global concern for celiac patients in food environments. Although some practices are positive, such as gluten labeling on processed foods in several countries, it is crucial to promote good practices in food services around the world. There are only few studies which did not show effective results in removing gluten from surfaces and utensils, besides sampling was limited, making it difficult to identify appropriate procedures to reduce cross-contamination. The variation in contamination in different kitchen environments also highlighted that celiac patients must continue paying attention to the methods used to prepare gluten-free foods. More research is needed, especially into methods of removing gluten from surfaces and utensils, to ensure food safety for celiac patients in many food environments.

The quality of this manuscript is good. It is well design and the result is interesting. However, there still have some issues need to improve.

1.      The Celiac Disease with inflammation should be analyzed.

2.      The gluten-free diet and Celiac Disease should be analyzed. Please refer this reference (Journal of Food Processing and Preservation,2021, 45(9),e15684.).

3.      The cytokines or inflammation status should be evaluated.

4.      The reason for combination and dose basis.

5.      The expression should be double checked.

6.      The reference should be updated in recent years.

Author Response

Dear Editor and Referees,

We would like to thank the referees and the editor for your valuable time during the review process and the insightful comments which helped us to improve the quality of the paper. We have appreciated all the considerations about our paper and we agreed with all of them.        

We have added the suggestions and corrections pointed out by the referees. Questions and suggestions are in black and answers are in blue.  If something is not exactly as it was suggested, please let us know and we can write it again.

Referee:3

Referee(s)' Comments to Author:                                                                                    

The quality of this manuscript is good. It is well design and the result is interesting. However, there still have some issues need to improve.

Response: Thanks for your gentle words, we have improved the paper, following our referees’ suggestions. All observations were accepted and corrected, besides all the authors agreed this version is much better than the first one.

1.The Celiac Disease with inflammation should be analyzed.             

2. The gluten-free diet and Celiac Disease should be analyzed. Please refer this reference (Journal of Food Processing and Preservation,2021, 45(9),e15684.).     

3. The cytokines or inflammation status should be evaluated.                   

4. The reason for combination and dose basis.                                         

Response: We have tried to follow your request covering all the topics highlighted and the references suggested for reading. We hope we have fulfilled your request by including the paragraphs below.

Celiac Disease (CD) is a chronic immune-mediated enteropathy of the small intestine, which manifests itself when exposure to dietary gluten occurs in genetically predisposed individuals [1]. The organ most affected by CD is the small intestine, impairing the ab-sorption of nutrients and causing clinical manifestations. Healthy villi are important for the proper digestion and absorption of nutrients [2, 3]. 

Gluten can reach the intestine by some mechanisms: the transcellular route, where gluten is endocytosed by lysosomes, which degrade it into smaller peptides [4], and the paracellular route, where tight junctions, which bind epithelial cells together, promote changes in cell permeability, allowing these gliadin peptides to enter the mucosa. An ex-ample is the regulation by zonulin, which is a protein synthesized by the intestine and liver and produced by the epithelial cells of individuals with CD, which causes this change in permeability [2]. Another mechanism is the adaptive immune response, which begins with the exposure of gluten peptides to CD4+ T cells in the intestinal mucosa, leading to the production of pro-inflammatory cytokines such as interferon-γ (IFN-γ). These cytokines stimulate T helper 1 cells to produce interleukins (IL-15 and IL-21 more specifically), activating CD8+ intraepithelial lymphocytes (IELs) and promoting intestinal damage. In addition, T-helper 2 cells induce the production of anti-gliadin, an-ti-transglutaminase and anti-endomysial antibodies, characteristic of CD. The increased presence of CD8+ IELs is also an important feature of CD, contributing to inflammation and damage to the intestinal mucosa [5].

5. The expression should be double checked.

Response: The expression was checked.

6. The reference should be updated in recent years.

Response: Following our referee suggestion, we have included more recent references as you can see below.

Sharma, N., Bhatia, S., Chunduri, V., Kaur, S., Sharma, S., Kapoor, P., ... & Garg, M. (2020). Pathogenesis of celiac disease and other gluten related disorders in wheat and strategies for mitigating them. Frontiers in Nutrition, 7, 6. https://doi.org/10.3389/fnut.2020.00006

Valitutti, F., & Fasano, A. (2019). Breaking down barriers: how understanding celiac disease pathogenesis informed the development of novel treatments. Digestive diseases and sciences, 64, 1748-1758. https://doi.org/10.1007/s10620-019-05646-y

Xiong, D., Xu, Q., Tian, L., Bai, J., Yang, L., Jia, J., ... & Duan, X. (2023). Mechanism of improving solubility and emulsifying properties of wheat gluten protein by pH cycling treatment and its application in powder oils. Food Hydrocolloids, 135, 108132. https://doi.org/10.1016/j.foodhyd.2022.108132

Round 2

Reviewer 1 Report

Comments and Suggestions for Authors

In the revised version, the Authors have addressed most of the relevant issues that I point out in my first review.

Comments on the Quality of English Language

A moderate editing of English is suggested

Author Response

 Dear reviewer, 

Please, accept the acknowledgments of our research group.
All researchers have highlighted your important and well-founded contributions to our paper. We also appreciate your kindness in carefully reading our paper.
We take this opportunity to point out that the paper has been re-evaluated by an expert in English as well.

Best Wishes, The authors

Reviewer 3 Report

Comments and Suggestions for Authors

The author did not respond the reviewer's comment point by point. Please update the comments.

Comments on the Quality of English Language

The author did not respond the reviewer's comment point by point. Please update the comments.

Author Response

Dear Editor and Referees,

             We would like to thank the referees and the editor for your valuable time during the review process and the insightful comments which helped us to improve the quality of the paper. We have appreciated all the considerations about our paper and we agreed with all of them.                                                                We have added the suggestions and corrections pointed out by the referee 3. Questions and suggestions are in black and answers :1st corrections are in blue, 2nd corrections in brown.

Referee:3

Referee(s)' Comments to Author:                                                                 

The quality of this manuscript is good. It is well design and the result is interesting. However, there still have some issues need to improve.

Response: Thanks for your gentle words, we have improved the paper, following our referees’ suggestions. All observations were accepted and corrected, besides all the authors agreed this version is much better than the first one.                                                        

          We have tried to follow your request covering all the topics highlighted and the references suggested for reading. We hope we have fulfilled your request by including the new paragraphs.

1. The Celiac Disease with inflammation should be analyzed.

Response: We thank our reviewer for his helpful advice. Following your request, we have added the paragraph below.

“Celiac Disease (CD) is a chronic immune-mediated enteropathy of the small intestine, which manifests when exposure to dietary gluten occurs in genetically predisposed individuals [1]. The organ most affected by CD is the small intestine, impairing the absorption of nutrients and causing clinical manifestations, caused by an inflammatory condition generated by the presence of gluten. Healthy villi are important for the proper digestion and absorption of nutrients [2, 3].”

2. The gluten-free diet and Celiac Disease should be analyzed. Please refer this reference (Journal of Food Processing and Preservation,2021, 45(9),e15684.).

Response: We agree with our reviewer about the importance and care of the gluten-free diet, besides part of our research group is formed by nutritionists. This multi-professional approach is very important in celiac disease. The gluten-free diet in this 2nd revision has been mainly highlighted in brown in the paragraphs: L.58-60; L.68-71; L.219- 223; L.224-225

We have included the reference, as advised by our reviewer.

            Li, Y., Shi, R., Qin, C., Zhang, Y., Liu, L., & Wu, Z. (2021). Gluten‐free and prebiotic oat bread: Optimization formulation by transglutaminase improvement dough structure. Journal of Food Processing and Preservation, 45(9), e15684.

3. The cytokines or inflammation status should be evaluated.  

Response: We agree with our reviewer and, as he requested, we have now included the paragraph below about the subject. L.49- L53

“Another mechanism is the adaptive immune response, which begins with the exposure of gluten peptides to CD4+ T cells in the intestinal mucosa, leading to the production of pro-inflammatory cytokines such as interferon-γ (IFN-γ). These cytokines stimulate T helper 1 cells to produce interleukins (IL-15 and IL-21 more specifically), activating CD8+ intraepithelial lymphocytes (IELs) and promoting intestinal damage.”

4. The reason for combination and dose basis.

Response: It was added a new paragraph L.68 - 72. In accordance with the standards of the Codex Alimentarius and the Food Safety, gluten-free products should not contain more than 20 ppm of gluten.

5. The expression should be double checked.

Response: The expression was checked.

6. The reference should be updated in recent years.

Response: Following our referee suggestion, we have included more recent references as you can see below.

Li, Y., Shi, R., Qin, C., Zhang, Y., Liu, L., & Wu, Z. (2021). Gluten‐free and prebiotic oat bread: Optimization formulation by transglutaminase improvement dough structure. Journal of Food Processing and Preservation, 45(9), e15684. https://doi.org/10.1111/jfpp.15684

Sharma, N., Bhatia, S., Chunduri, V., Kaur, S., Sharma, S., Kapoor, P., ... & Garg, M. (2020). Pathogenesis of celiac disease and other gluten related disorders in wheat and strategies for mitigating them. Frontiers in Nutrition, 7, 6. https://doi.org/10.3389/fnut.2020.00006

Valitutti, F., & Fasano, A. (2019). Breaking down barriers: how understanding celiac disease pathogenesis informed the development of novel treatments. Digestive diseases and sciences, 64, 1748-1758. https://doi.org/10.1007/s10620-019-05646-y

Xiong, D., Xu, Q., Tian, L., Bai, J., Yang, L., Jia, J., ... & Duan, X. (2023). Mechanism of improving solubility and emulsifying properties of wheat gluten protein by pH cycling treatment and its application in powder oils. Food Hydrocolloids, 135, 108132. https://doi.org/10.1016/j.foodhyd.2022.108132

Round 3

Reviewer 3 Report

Comments and Suggestions for Authors

It is better to check the Lanage expression issue.

Comments on the Quality of English Language

It is better to check the Lanage expression issue.